psychology

questionable research practices, statistical reporting, statistical power, effect size, academic theses, *p*-value

**Author for correspondence:**
Jerome Olsen
e-mail: jerome.olsen@univie.ac.at

# Research practices and statistical reporting quality in 250 economic psychology master's theses: a meta-research investigation

Jerome Olsen[1], Johanna Mosen[1], Martin Voracek[2] and Erich Kirchler[1]

[1]Faculty of Psychology, Department of Applied Psychology: Work, Education and Economy, University of Vienna, Universitaetsstrasse 7, 1010 Vienna, Austria
[2]Faculty of Psychology, Department of Basic Psychological Research and Research Methods, University of Vienna, Liebiggasse 5, 1010 Vienna, Austria

JO, 0000-0002-7812-7322; MV, 0000-0001-6109-6155;
EK, 0000-0003-4731-1650

The replicability of research findings has recently been disputed across multiple scientific disciplines. In constructive reaction, the research culture in psychology is facing fundamental changes, but investigations of research practices that led to these improvements have almost exclusively focused on academic researchers. By contrast, we investigated the statistical reporting quality and selected indicators of questionable research practices (QRPs) in psychology students' master's theses. In a total of 250 theses, we investigated utilization and magnitude of standardized effect sizes, along with statistical power, the consistency and completeness of reported results, and possible indications of *p*-hacking and further testing. Effect sizes were reported for 36% of focal tests (median $r = 0.19$), and only a single formal power analysis was reported for sample size determination (median observed power $1 - \beta = 0.67$). *Statcheck* revealed inconsistent *p*-values in 18% of cases, while 2% led to decision errors. There were no clear indications of *p*-hacking or further testing. We discuss our findings in the light of promoting open science standards in teaching and student supervision.

## 1. Introduction

The replicability of research findings has recently been disputed across multiple scientific disciplines (e.g. [1–4]). The main reasons for this replicability debate are seen in the ubiquity of

underpowered studies [5–7], in combination with questionable research practices (QRPs; e.g. *p*-hacking, HARKing (hypothesizing after the results are known), dropping conditions; [8–10]) and publication bias [11]. Consequences of these phenomena are a large proportion of false-positive findings, biased effect size estimates and distorted meta-analytic conclusions [12,13].

In constructive reaction, the research culture in psychology is facing fundamental changes [14,15]. These include practices such as open data and materials, along with preregistration of design, hypotheses and analysis plan [16–19]. These measures ensure transparency of the entire research process on all stages and aim at facilitating reliable and reproducible research findings. Additionally, there is a reversion to practices which traditionally have been seen as vital for null-hypothesis significance testing, but, to some extent, have been neglected, such as *a priori* power analysis and the interpretation of effect sizes [20,21]. All of these recommended practices are not yet fully adopted by all [22], but the service infrastructure has been laid out (e.g. Open Science Framework), and an increase in practices has been documented among researchers [23,24], as well as in scientific journals [18,25,26]. Furthermore, researchers increasingly are organizing themselves in large-scale collaborative networks to overcome limitations such as small sample size, non-generalizable samples and specific measurement settings (e.g. [27–29]).

Investigations into research practices and scientific output that led to these improvements have almost exclusively focused on academic researchers (e.g. [11,30]). Extending this perspective, we believe that exploring psychology students' statistical reporting quality and indications of QRPs would be just as important, as their behaviour reflects our quality of teaching and they are tomorrows' early-career researchers in psychological science. We are aware of a single recent study that investigated psychology students' self-reported behaviour and attitudes regarding QRPs [31]. These survey results revealed an average QRP self-admission rate of 9% over 11 different practices, while behaviours such as selective reporting of studies that worked,[1] data exclusions after looking at the data and HARKing had rates of 15% and higher. Statements that significant results would lead to better grades or would be an indication of good science were generally not approved, whereas reporting effect sizes and conducting power analysis was endorsed by 69% and 35%, respectively.

These reported QRP self-admission rates of students are lower than those of academic researchers [9,10].[2] However, Krishna & Peter [31] discuss the possibility that students' self-admission rates might underestimate actual QRP prevalences in student theses. It therefore remains unclear to what extent students' actual behaviour reflects sacrifices of scientific rigor to achieve significant results, considerations about statistical power and correct statistical reporting when writing their final thesis. The present study sought to fill this gap by descriptively analysing master's theses in psychology supervised at the University of Vienna, Austria, between 2000 and 2016. We will discuss our findings in the light of promoting open science standards in teaching and student supervision.

## 1.1. Reasons for questionable research practices

A key piece of career advice for academic researchers is to publish research, ideally a lot and in reputable journals [32,33], as publication output is usually a prerequisite to receiving a PhD, and further is linked to hiring, salary, tenure and grant decisions [34,35]. Additionally, publication numbers are used as a criterion to evaluate and rank departments and universities [36–38]. In the presence of publication bias, this can create an imbalance between the need to produce a publishable paper and conducting accurate research, and is often argued to be a source of QRPs to increase the publishability of research findings [39,40]. Clearly, this 'publish-or-perish' imperative is unlikely to apply to master's students who preponderantly do not pursue a publication of their theses in a scientific journal or, at least, are not required to publish for getting their master's degree.

[1]Krishna & Peter [31] were surprised by the high self-admission rate of this particular behaviour and discuss the possibility that students might have interpreted this question differently from researchers. As it is less common for students to run multiple studies, participating students might have interpreted the item as referring to selectively citing papers that worked, rather than to a series of empirical studies conducted by oneself.

[2]Here, we compared the *self-admission* rates reported in the three publications. In their paper, Krishna & Peter [31] state that 'comparing our results to Fiedler & Schwarz' [10] estimated prevalence rates of individual QRPs, most QRPs have similar self-reported prevalences'. However, they arrive at this conclusion by comparing students' *self-admission* rates with Fiedler & Schwarz' [10] *prevalence* rates. We believe it is more informative to compare the same estimates between the three studies, namely the *self-admission* rates. In this comparison, students' self-admission rates are always lowest (except for 'falsifying data', however, where all self-admission rates were very low in absolute terms).

Another mechanism that frequently is mentioned as a source of QRPs is the tendency to interpret outcomes in one's own favour, namely motivated reasoning [41] and confirmation bias [42]. Falsification is the core principle of critical rationalism [43], but if, after data collection, researchers are given the flexibility to choose between different types of analyses, decide for data exclusions or report only those outcomes that yield a significant effect, motivated reasoning and confirmation bias could hinder researchers from giving up their personal conviction of a phenomenon [44,45].

Unconscious motives like motivated reasoning could also play a role among master's students, which might be a relevant similarity between academic researchers and students. Completing the master's thesis is an important milestone for students. They have to invest considerable resources to answer a research question on their own. On the one hand, it is conceivable that such a situation motivates students to show their ability to conduct rigorous scientific practice; on the other hand, students might also develop a sense of wanting to find support for what is their own perceived truth or idea.

Therefore, the current research might provide a preliminary indication on whether QRPs are more likely to stem from our statistical training and socialization as psychologists (i.e. we observe indications of QRPs among students' master's theses), or rather from a 'publish-or-perish' culture (i.e. we do not observe indications of QRPs in master's theses). If we observe a prevalence of QRPs that is similar to those of publishing researchers, this could suggest that similar mechanisms are at work between these two groups. It would express a common belief in the need to produce statistically significant findings and could be interpreted to show that mechanisms like motivated reasoning mainly cause high QRP rates as they are also observable in circumstances where publication pressure should not be influential. If, on the other hand, we observe no substantial QRPs among students, this would suggest that the prevalence of QRPs among academic researchers is more likely to stem from structural characteristics that only apply to researchers and could be interpreted to show that QRPs are more likely to stem from a 'publish-or-perish' culture. We will not be able to give a final answer to this question based on our data, but we might be able to give a cautious indication.

## 1.2. Formal thesis supervision and submission criteria at the University of Vienna

The Faculty of Psychology at the University of Vienna offers bachelor and master studies in psychology. Students in the master programme can apply for master's thesis supervision at three different departments within the Faculty of Psychology. Thematic areas cover (i) cognitive, emotional and motivational processes and their biological foundations, (ii) development, education and learning over the life course, (iii) health, strain, coping and social inclusion, (iv) work, society and the economy, and (v) research methodology and synthesis. The present study focused on theses in economic psychology, only. Thesis topics can either be proposed by students or are offered by supervisors. The majority of projects are empirical, while writing theoretical conceptual, or methodology theses is possible as well. Master's theses must adhere to the formal requirements of the *Publication Manual of the American Psychological Association* [46] or the slightly adapted German version thereof [47]. There are no formal requirements on thesis length. Submitted theses can be written in the national language (German) or in English.

While most procedural steps of conducting research as a student are similar to those of academic researchers, there are also some important differences. First, students usually do not pursue a publication of their findings in a scientific journal. Some supervisors may invite students to submit a revised version to a journal after the supervision is concluded, but it is usually not expected that manuscripts are submission ready. Second, students have less experience in statistics, methodology and scientific thinking at large. Third, students must submit their thesis before a fixed date each month, in order to be eligible to defend their thesis in the following month. Most students aim for a particular month to graduate in, which can create substantial time pressure. Finally, students have only little access to financial means to pay for incentives. Studies are therefore often characterized by hypothetical rather than real decision paradigms and by the use of convenience samples.

Each year 485 bachelor's students are admitted to the programme who, after successful completion, can continue with the master's programme. Most graduates of the master's programme find positions in a clinical context (55%), followed by jobs in consulting (11%), training (9%) and leadership (6%). The average student does not pursue (or find) a career in academia, as only 4% of students occupy a research position after graduation [48].

Each supervisor holds their own master's thesis seminar where students generally are expected to present their research projects at two stages: once after planning the study (presentation of theoretical background, research questions and hypotheses, and methods) and once after data collection

(presentation of results and discussion). We believe this structure already reduces the likelihood for students to change their hypothesis after data collection, as they are required to present these prior to data collection. However, it does not formally hinder students from *p*-hacking, as they usually only state their hypotheses, but seldom present *a priori* power analysis or a detailed analysis plan. Therefore, degrees of freedom remain high when it comes to selective reporting, further testing, including covariates, excluding cases, composing variables or choosing between data-analytic approaches.

As a reaction to the reproducibility discussion in psychological science, we have improved the structure of the master's thesis seminars (held by authors E.K. and J.O.) continually over the past years. Master's thesis candidates now are required to present a justification of their sample size and their analysis plan in the first presentation. Furthermore, this presentation is archived on an internal server and is regarded as a preregistration, while additional preregistration via platforms such as OSF is recommended.[3] When presenting for the second time, after data collection, students may only interpret those hypotheses inferentially that were formulated *a priori*, and must designate all additional analysis as exploratory. Hence, the seminar now takes on the form of an oral registered report, for which the supervisor and the further seminar attendees act as reviewers at stage 1 (first presentation). To facilitate the required theoretical and practical knowledge, we introduce the topic open science as a state-of-the-art research requirement at the beginning of each teaching term. Regardless of these improvements, we are unaware of the statistical quality of master's theses in years prior to these improvements.

## 1.3. Research questions

### 1.3.1. Research question 1: Do students report standardized effect sizes, and what is the average effect size of focal effects?

As researchers, we should not only be interested in whether the null hypothesis can be rejected, but also in the magnitude of an effect. Lakens [49] argues that effect sizes are the most important outcome of empirical studies. First, they allow science communication in a standardized metric; second, they are directly comparable, which is especially important for meta-analysis; and third, they guide sample size decision in power analysis for future studies.

Among others, the importance of reporting effect sizes is also emphasized in the *Publication Manual of the American Psychological Association* ([46], (p. 33)). But do publishing researchers follow these widely recommended standards? A comprehensive review (information on over 6000 articles) found that only 38% of articles published in 1990–2007 reported effect sizes [50]. However, it is noteworthy that the ratio of articles reporting effect sizes increased over time reaching 67% by 2007 in non-clinical fields.

As to the average size of effects reported in psychological science, a large meta-meta-analysis, summarizing 100 years of social psychological research, found a mean effect size of $r = 0.21$ [51]. The authors do consider that publication bias might have inflated this estimate, but a sensitivity analysis of meta-analyses that also included unpublished findings (grey literature) attested only a small upward bias. Pre-registered replication studies are less prone to the criticism of potentially inflated effect estimates. The Reproducibility Project: Psychology (RP:P; [1,52]) found a mean effect size of $r = 0.20$ in replications of 100 studies published in three high-ranked psychology journals. The original studies had a mean effect size of $r = 0.40$. Similarly, a replication project of social science studies published in *Nature* and *Science* found a mean effect of $r = 0.25$, whereas the original studies had $r = 0.46$ [53].

Here, we investigated the proportion of students reporting standardized effect sizes for their focal test result in their master's thesis and calculated the average observed effect size. According to students' self-reports, 69% refer to effect sizes in their final academic thesis [31].

### 1.3.2. Research question 2: Do students report *a priori* power analysis, and what is the average observed power of focal effects?

Power is the probability that a statistical test correctly rejects the null hypothesis, given the effect is non-null. Conventionally, a power of 0.80 is an accepted minimum [54], whereas higher power figures generally are recommendable, as long as this practically is feasible.

---

[3]As a next step of improvement, we are currently planning to make it mandatory to submit a completed as-predicted preregistration.

Historically, there has been repeated emphasis on the importance of power analysis, along with criticism of researchers' persisting neglect to report and conduct power analysis [55–57]. Changes in this domain seem to progress only slowly, and estimates of the proportions of articles reporting power analyses are in the range of 3–5% [11,50,58].

This neglect seems to reflect researchers' (incorrect) intuitions and assumptions about required sample sizes to detect effects of a certain magnitude. When researchers were asked about acceptable sample sizes to study an expected effect size, the resulting power with their indicated sample size would only have been around 0.40. When asked to estimate the power of a research design based on $\alpha$, effect size and $N$, respondents overestimated analytic power for designs involving small effects, whereas they underestimated power figures in the case of large effects [5]. In combination, this suggests that researchers' intuitions favour underpowered studies, especially in the case of small effects, which are more common in psychology than large effects (see Research question 1). Accordingly, average power in psychology has retrospectively been estimated to have hovered merely around 0.35 [12].

At the same time, it has been argued that even if researchers neglect power considerations, students might rely on them, as it should be part of their formal statistics training [59]. Students' self-reports provide a prevalence estimate of 35% for their final thesis [31]. We investigated the proportion of students actually reporting power considerations (formal power analysis, or rule of thumb) and the average observed power of focal effects.

### 1.3.3. Research question 3: Which statistical reporting errors can be observed?

It is crucial that reported statistics in scientific papers are correct. Nevertheless, many research papers contain statistical reporting errors [60,61]. These can be the result of simple typos or misreading statistical output, whereas in some cases, they might express deliberate misreporting in favour of researchers' expectations. *Statcheck* is an R package that can be used to check whether reported $p$-values are consistent with reported test statistics and degrees of freedom (d.f.) [62]. Applying this tool to articles published in six prominent psychology journals between 1985 and 2013 revealed that 12.9% of articles contained at least one gross inconsistency (i.e. an error changing the statistical conclusion) and 49.6% at least one general inconsistency (i.e. an error not changing the statistical conclusion), which amounts to 1.4% and 9.7% of all checked $p$-values, respectively [63]. We checked whether students report consistent $p$-values.

Besides focusing on inconsistencies of $p$-values, one can look at the completeness of reported statistics. Bakker & Wicherts [60] found that either $p$-values, d.f., or test statistics were missing in around 31% of cases in high-impact journals. Hence, we additionally investigated whether students reported complete statistical results.

### 1.3.4. Research question 4: Does the $p$-value distribution indicate $p$-hacking?

The distribution of $p$-values is frequently used to investigate the presence of $p$-hacking in sets of multiple studies. Under the null hypothesis, $p$-values are uniformly distributed. If there is a true effect, $p$-values form right-skewed distributions, i.e. smaller $p$-values become more likely than larger ones. As some proposed hypotheses are true, whereas others are false, the distribution of $p$-values should be a mix of a uniform distribution and a monotonically decreasing right-skewed distribution [64,65].

Most QRPs ultimately influence the $p$-value so that a significant (i.e. usually a $p$-value below the conventional significant threshold of 0.05) can be reported, turning the finding into a false-positive. This can be achieved by analysing data flexibly with the aim of lowering the $p$-value (e.g. removing outliers, transforming data, trying different statistical tools). One way of observing these occurrences is by inspecting the distribution of $p$-values. $p$-hacking is claimed to be evident if there is a bump of $p$-values just below 0.05. In fact, a number of studies have suggested that there is such a bump in the distribution of $p$-values of published articles [66–68].

However, some authors have argued that it is not appropriate to draw general conclusions about the prevalence of $p$-hacking merely based on the distribution of $p$-values [69–71]. According to Lakens [69], one must consider whether there is between-study heterogeneity with regard to average power, the ratio of true positives to false positives and the $\alpha$ level. For instance, if there are two studies investigating the same effect, but one is underpowered, the $p$-value of the underpowered study will be larger. While these three attributes should influence the distributional steepness (versus flatness), the distribution itself

should be monotonic. However, one additional point to consider is whether there is publication bias, which would cause a systematic absence of *p*-values above the significance threshold and therefore would result in a non-monotonic distribution of *p*-values. Importantly, publication bias is very unlikely to apply in the case of students' master's theses. Therefore, we expect a monotonic *p*-value distribution if *p*-hacking is absent.

### 1.3.5. Research question 5: Do sample size and effect size correlate negatively?

If researchers conducted power analysis, they would run studies with sample sizes optimized to detect effects of expected magnitude. They would use small samples when expected effects are large, and large samples when expected effects are small, thus resulting in a negative correlation between sample size and effect size [59,72]. However, as we have seen earlier, researchers seldom report power considerations [50].

Therefore, in meta-analysis, a negative correlation between sample size and effect size is usually interpreted as a sign of publication bias [73,74], where only those studies are reported in the literature that yield nominal significance. In such a situation, studies investigating small effects with small samples (non-significant findings) are missing in the literature, and the remaining significant findings produce a negative correlation between sample size and effect size.

But as stated earlier, in the case of students, it is very unlikely that there is publication bias. Here, one plausible explanation for such a negative correlation would be further testing. Assuming that students have a self-interest to produce significant results, they could start collecting data, check whether their result is significant and, if not, continue to collect data. This would systematically bias small effects to be investigated with larger samples in the absence of clear stopping rules.

## 2. Method

### 2.1. Inclusion criteria and sampling

A master's thesis was eligible for inclusion if it was supervised as part of author E.K.'s master's thesis seminar (Professor of Economic Psychology at the University of Vienna), reported a null-hypothesis significance test of a focal hypothesis that was not tested in another thesis using the same data and was submitted between 2000 and 2016. We had full access to a total of 329 master's theses in economic psychology that were completed within this time span. After excluding meta-analytic, non-empirical, qualitative, descriptive and exploratory work, as well as master's theses of different students who tested identical focal hypotheses using the same data, the final number of eligible master's theses was 250.

### 2.2. Coding scheme

The coding scheme closely followed strategies employed in prior related meta-research [11,30]. The strategy was to identify the study design and focal hypothesis, followed by identifying all relevant statistical information of the respective inferential test. Table 1 presents an overview of the coded information by category. Coding was performed by authors J.O. and J.M. and an undergraduate research assistant.

### 2.3. Analysis plan

Research question 1: We investigated the proportion of students reporting standardized effect sizes for their focal test result in their master's thesis and calculated the average observed effect size.

Research question 2: We first investigated the proportion of students actually reporting power considerations (formal power analysis, or rule of thumb), followed by estimating the average observed post hoc power of focal effects. One frequently discussed limitation of post hoc power estimations for sets of studies is that they are upwardly biased to suggest higher power when there is publication bias (e.g. [24]). However, in our case, the opposite might be the case, as student theses probably include true negatives. Think of a study with $N = 500$ individuals that finds a correlation of $r = 0.03$. Let us assume this is a true negative and the non-zero estimate is a result of sampling variance. The observed post hoc power would only be 10%. However, consider using the same sample size for a true effect of $r = 0.17$. Post hoc power would now be 97%. This means a post hoc power analysis of a

**Table 1.** Coded characteristics.

| category | coded information |
| --- | --- |
| formal characteristics | year |
| | length |
| | thesis topic |
| classification | type: |
| | 'non-empirical', 'qualitative', 'descriptive', 'exploratory' (excluded) |
| | 'inferential' (included) |
| hypothesis | focal hypothesis (if there was no clear focal hypothesis, the first hypothesis was used.) |
| variables | dependent variable |
| | independent variable/s |
| sample | sample size criterion: 'none', 'power analysis', 'rule-of-thumb' |
| | type of sample: 'student', 'non-student, no further specification', 'employed', 'self-employed', 'children' |
| | sample size |
| | mean age and standard deviation |
| analysis | test type: |
| | 'test of categorical data ($\chi^2$)' |
| | 'test of mean difference ($t$-test)' |
| | 'analysis of variance ($F$-test)' |
| | 'correlation' |
| | 'linear regression' |
| | 'non-parametric test' |
| | 'complex analysis' |
| | test statistics (depending on method): |
| | reported test value, d.f., cell means and standard deviations, independent/dependent |
| | did the model include covariates: 'yes'/'no' |
| $p$-value | reported as: |
| | 'exact, 3 decimals' |
| | 'exact, 2 decimals' |
| | 'significant threshold (e.g. <0.05)' |
| | 'non-significant threshold (e.g. >0.05)' |
| | 'missing' |
| | reported $p$-value |
| effect size | was an effect size reported for the focal hypothesis test: 'yes' / 'no' |
| | were effect sizes generally reported: 'yes'/'no' |
| | type of effect size: '$r$', 'Cohen's $d$', 'Glass's $\Delta$', 'Hedges' G', '$f^2$', '$\eta^2$', '$R^2$', 'standardized $\beta$', 'unstandardized $\beta$', 'Cramer's $V$', 'odds ratio' |
| | reported effect size |
| reporting errors | reporting errors: '$p$-value missing', 'test value missing', 'd.f. missing', 's.d. missing' |
| data use | data were also used in other thesis: 'yes'/'no' |
| | focal hypothesis was the same: 'yes'/'no' |
| exploration | further testing beyond hypothesis testing: 'yes'/'no' |
| | exploration was reported in designated section: 'yes'/'no' |

body of literature that includes true negatives could easily underestimate true power.[4] Given the risk of a false power estimate based on this approach, we furthermore estimated the hypothetical power that the used sample sizes would have for different conventional effect sizes.

Research question 3: To evaluate whether reported $p$-values are consistent with reported test statistics and d.f., we used the R package *statcheck* (v. 1.3.0; [62]). Furthermore, we determined the proportion of students failing to either report $p$-values, d.f., or test statistics as a measure of reporting completeness.

Research question 4: We first visually inspected the $p$-value distribution followed by conducting a $p$-curve analysis. P-curve is a tool explicitly designed for a situation without access to non-significant results [30,80] which is not the case in the present study. However, focusing only on the reported significant results, it provides a test for skew of $p$-values within the significance range between 0 and 0.05. As outlined above, if there is a true effect, $p$-values form right-skewed distributions. If there was severe $p$-hacking in the theses, $p$-curve would attest a lack of evidential value, i.e. a left-skewed $p$-curve (only $p$-hacked results of null effects paired with low power, which is very unlikely to be the case), or a combination of right-skewed with a left-skewed $p$-curve (resulting from a mix of true effects and $p$-hacked null effects).

Furthermore, to assess whether there is a bump in the $p$-value distribution that could signify $p$-hacking, we calculated calliper tests [81,82]. This proportion test compares the frequencies of two intervals of the same distribution.[5] We would assume that in the presence of $p$-hacking, $p$-values in the interval 0.045–0.0499 are observed more often than values in the intervals just below (i.e. 0.040–0.0449) and above (i.e. 0.050–0.0549).

Research question 5: To test whether there is a bias for small effects to be investigated with larger samples, which could express further testing, we correlated sample size with effect size and visually inspected the scatterplot. A negative correlation could express an absence of studies investigating small effects with small sample sizes.

Data and R code for this study can be retrieved from https://osf.io/jwaqv/.

# 3. Results

## 3.1. Research question 1: Do students report standardized effect sizes, and what is the average effect size of focal effects?

In 90 cases (36% of all theses), students reported a standardized effect size for their focal hypothesis test. Of these, the most frequently reported metrics were standardized regression coefficients, i.e. $\beta$ (36%), $\eta_P^2$ (31%) and Pearson's $r$ values (21%). In some theses, standardized effect sizes were generally reported, but omitted, if the focal test was not statistically significant. Therefore, the proportion of standardized effect sizes reported in general was 44%.

Whenever sufficient information was available (e.g. cell $M$ and s.d. values, test statistic and d.f.), reported effect sizes of focal tests were checked and missing effect sizes were calculated. The resulting number of standardized effect sizes was 223 (89%), which were all converted to Pearson's $r$ values. Figure 1 depicts a violin plot of all resulting effect sizes. The mean effect size was $r = 0.24$, 95% $CI$ [0.21, 0.26]. Due to considerable skew and outliers in the upper end, the median seems more informative which was $r = 0.19$, 95% $CI$ [0.16, 0.22].

## 3.2. Research question 2: Do students report *a priori* power analysis, and what is the average observed power of focal effects?

Of all 250 inspected master's theses, only a single student reported a formal power analysis. Another five reported a rule of thumb. The average sample size was $M = 222$ (s.d. = 284, range from 21 to 3242), $Mdn = 157$ ($IQR = 140$).

---

[4]Furthermore, it has often been argued not to conduct post hoc power analysis at all. The criticism concerns situations where only a single study is considered [75–77]. The observed power is based on the observed effect size (together with the used sample size and $\alpha$-level). The problem is that the observed effect size of a single study is often a noisy estimate. However, it is argued that for sets of studies, the median post hoc power can be a justified estimate, as the aggregation of effect sizes reduces the sampling error across studies [78,79].

[5]Note that the original test is used to attest publication bias by assessing whether there are more significant than non-significant test results. For this purpose, the proportion of studies with $z$ values below the critical $z$ value are compared against those above the critical $z$ value. In this study, we used intervals for $p$-values.

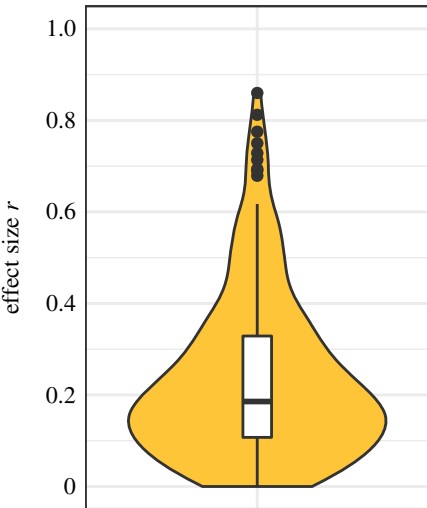

**Figure 1.** Violin plot of effect sizes.

Figure 2 depicts two different power plots. Figure 2*a* shows the observed power of theses, which is the post hoc power calculated for each study's sample size with its corresponding observed effect size. The bimodal distribution indicates that some studies were adequately powered, whereas others either had substantially too little power or studied true negatives. The resulting estimated median post hoc power was $1 - \beta = 0.67$, 95% CI [0.54, 0.78].

Another way of describing the power of a set of studies is to calculate the power each sample size would have for different effect sizes, and then take the median of these estimates for each effect size. Figure 2*b* depicts the resulting power curve that illustrates the samples' median power for the full range of possible Pearson's *r* effect sizes. Starting with $1 - \beta = 0.05$ for a null effect (equal to the $\alpha$-error probability), the median power increases with increasing effect size until reaching a power of $1 - \beta = 1$ at around $r = 0.40$. Note that, based on this approach, the median power for $r = 0.19$ (median effect size of focal effects) would be $1 - \beta = 0.65$, 95% CI [0.59, 0.71], which is similar to the observed post hoc power estimate reported above. As for conventional effect sizes, the sample sizes used would have adequate power for large effects, $r = 0.30$, $1 - \beta = 0.97$, 95% CI [0.95, 0.98], nearly acceptable power for medium effects, $r = 0.20$, $1 - \beta = 0.71$, 95% CI [0.66, 0.77], but clearly too little power for small effects, $r = 0.10$, $1 - \beta = 0.24$, 95% CI [0.22, 0.27]. Note also that power reaches the recommended minimum power threshold of $1 - \beta = 0.80$, once effects are $r = 0.22$.

## 3.3. Research question 3: Which statistical reporting errors can be observed?

Using the R package *statcheck*, we were able to recalculate 174 *p*-values based on reported test statistics and d.f. Overall, 32 of the 174 focal test results contained inconsistencies (18%), whereas 4 of these (2%) led to changes in the conclusion of the hypothesis test. In three of these four cases, the null hypothesis was claimed to be rejected, albeit the correct *p*-value (based on the reported test value and d.f.) actually was larger than 0.05.

As for the completeness of reported statistics, either *p*-values, d.f., or test statistics were missing in 14% of theses. This number was influenced most by missing d.f. (9%).

## 3.4. Research question 4: Does the *p*-value distribution indicate *p*-hacking?

In about half of the theses, reported *p*-values were below 0.05 (53%). Visual inspection of the *p*-value distribution (figure 3) revealed a clear right-skew with the majority of significant *p*-values falling below 0.01, as would be expected for a body of studies containing true effects. Furthermore, non-significant *p*-values are distributed along the entire possible range of values. The right-skew is also attested by a *p*-curve analysis of 130 eligible significant study results (figure S1 in the OSF repository), $p_{half} < 0.001$ and $p_{full} < 0.001$.

Despite the clear visual evidence and the fact that there were very few *p*-values just around the 0.05 significance threshold, we calculated a caliper test [81,82], to assess whether there is a bump in the *p*-value distribution that could signify *p*-hacking. One-sided binominal tests did not reveal more *p*-values in the interval 0.045–0.0499 than in the interval below (i.e. 0.040–0.0449), $p = 0.344$, or above

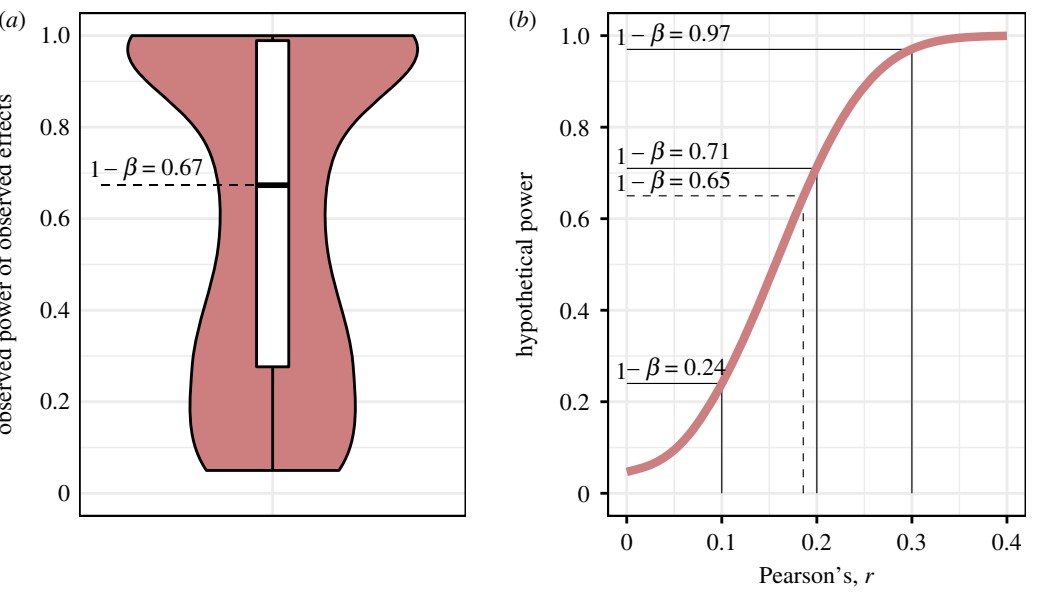

**Figure 2.** Power plots. (a) Observed post hoc power of theses. (b) Median hypothetical power for various effect sizes from $r = 0$ to $r = 1$, i.e. calculated based on each occurring sample size and all possible effect sizes. Solid lines indicate the samples' power for conventional effect sizes of $r = 0.10$, $r = 0.20$ and $r = 0.30$; the dotted line represents the power for the median observed effect estimate of $r = 0.19$.

(i.e. 0.050–0.0549), $p = 0.746$. In conclusion, there was no indication of a bump in the $p$-value distribution that would suggest considerable presence of $p$-hacking.

## 3.5. Research question 5: Do sample size and effect size correlate negatively?

Correlating effect size and sample size for the 223 cases where standardized effect sizes were available or computable yielded a Spearman rank-order correlation of $r_s = -0.10$, 95% CI [−0.23, 0.04], $p = 0.142$. See figure 4 for a scatterplot of effect size against the logarithmically transformed sample size. Hence, there was no clear indication of a bias taking the form that smaller effects would result from larger samples and vice versa.

## 3.6. Exploration of changes over time

To explore whether reporting quality changed over time, we correlated the year a thesis was graded with the share of reported focal effect sizes and the completeness of reported statistics (i.e. either $p$-values, d.f., or test statistics missing). Over the years, the number of reported standardized effect sizes of focal effects increased, $r(248) = 0.17$, 95% CI [0.04, 0.29] (from 28% until 2008 to 47% in years after 2008). This becomes even more evident when looking at the increase in the use of standardized effect sizes in general (irrespective of the focal test result; from 31% until 2008 to 61% in years after 2008), where the correlation with years was $r(248) = 0.26$, 95% CI [0.15, 0.38]. $p$-value inconsistencies did not change over time $r(172) = -0.04$, 95% CI [−0.19, 0.11], nor did the completeness of reported statistics, $r(248) = 0.06$, 95% CI [−0.06, 0.18].

Furthermore, we explored the correlations of year with sample size, focal effect size and observed power. The sample size was not related to time, $r(248) = 0.01$, 95% CI [−0.11, 0.14], but investigated focal effects seemed to become smaller over time, $r(224) = -0.15$, 95% CI [−0.28, −0.02] (from $Mdn = 0.21$ until 2008 to $Mdn = 0.17$ in years after 2008). In combination, this resulted in a decreasing trend in observed power, $r(224) = -0.09$, 95% CI [−0.22, 0.04] (from $Mdn = 0.76$ until 2008 to $Mdn = 0.54$ in years after 2008). For an illustration of these relationships, see figure S2 in the OSF repository.

## 4. Discussion

We investigated selected indicators of QRPs and statistical reporting quality in 250 psychology students' master's theses supervised at the University of Vienna between 2000 and 2016. Most importantly, there

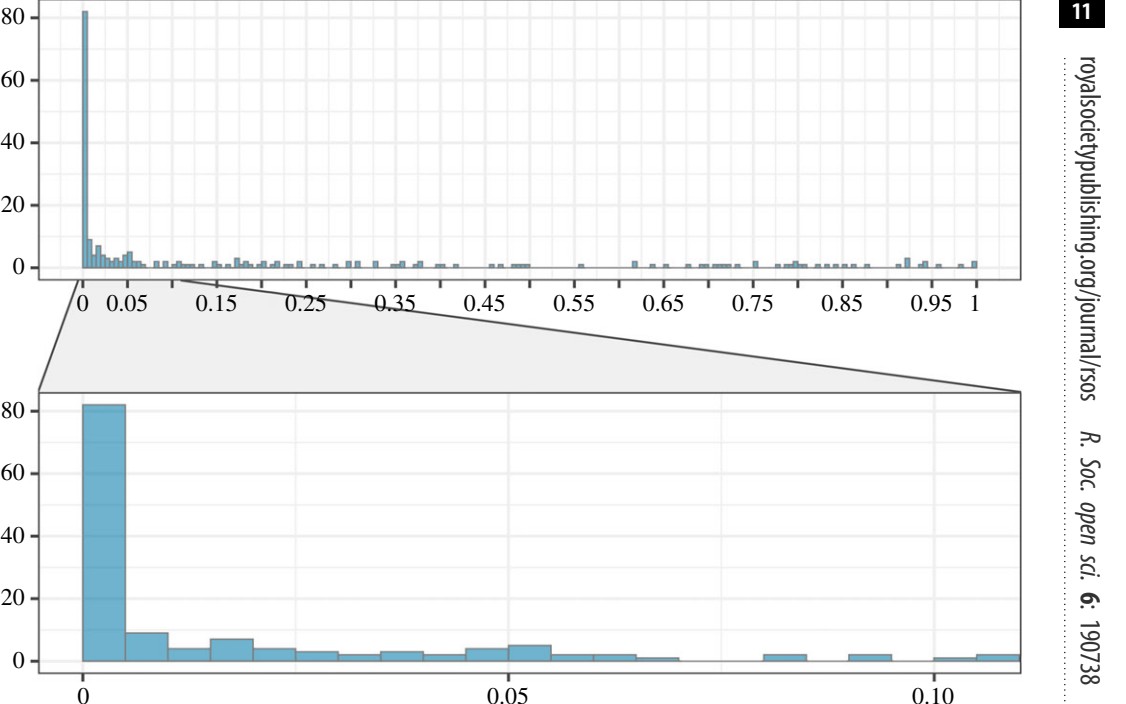

**Figure 3.** Distribution of 226 *p*-values that were recalculated with *statcheck* or, if not possible, reported as exact values. The non-included 24 *p*-values were either not reported ($n = 4$) or reported as smaller or larger than a common threshold (e.g. '<0.05', $n = 20$), and could not be recalculated.

were no signs of QRPs in terms of severe *p*-hacking or further testing. Regarding the reporting quality, we find that in terms of consistency of *p*-values, completeness of statistical reporting and relying on standardized effect size metrics there is room for improvements—which are already observable for reporting effect sizes, as revealed by our exploration of changes over the years.[6] One major shortcoming was the neglect of power considerations in nearly all theses. The share of students relying on a power analysis (0.4%) was much lower than self-reports by German psychology students (35%; Krishna & Peter [31]). However, the resulting hypothetical power of the used sample sizes was, while not perfect, decent to detect effect sizes of magnitudes common in psychological research.

## 4.1. Comparisons between students and publishing researchers

Our results reveal certain similarities between students and publishing researchers, but also some clear differences. One major difference is the absence of substantial QRPs among students. This makes it more likely that observed QRPs in the published literature stem from circumstances that are exclusive to researchers and do not apply to students. While we cannot fully answer this question, as our sample is not representative of the entire field, this observation is in support of the common narrative that publication pressure in a 'publish-or-perish' environment created an imbalance between the need to produce a publishable paper and conducting rigorous research (before outcome independent publishing became a realistic option). Another structural difference between students and researchers is that students have less experience in conducting empirical research. Hence, an alternative, but rather pessimistic, explanation for the absence of severe QRPs among students could be that they simply do not know how to exploit flexibility in data analysis.

In comparison with academic researchers, the included student theses are characterized by almost twice as many inconsistent *p*-values [63]. This could stem from less experience in conducting empirical research and less established work routines. Recall that the majority of inconsistencies were reporting errors that did not change the statistical conclusions. This suggests that the inconsistencies are more likely to be careless errors than deliberate misreportings in terms of *p*-hacking. Counter to

---

[6]Note that the share of reported effect sizes in general in later years (62%) was close to student self-reports from Germany (69%; [31]).

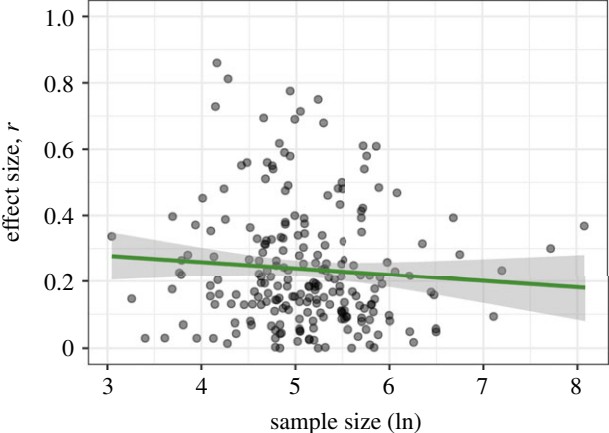

**Figure 4.** Correlation of sample size and effect size *r*.

this argumentation, the theses contained half as many missing key statistics than is the case in published articles (*p*-values, d.f. or test statistic; [60]).

The average effect size (around *r* = 0.20) corresponds to figures reported in broad replication projects [1,53] and meta-meta-analysis in social psychology [51]. This could suggest that students are as good as publishing researchers in finding effects over a range of different research paradigms. In the context of replication studies, some authors have voiced the concern that failures to successfully replicate an effect could be due to unmeasured differences between original and replication procedures (e.g. [83]). Despite that investigated theses were not replication studies *per se*, our finding could be interpreted to show that experimenters' expertise is unlikely to explain large variability in effect size between studies.

One rather concerning similarity is the neglect of power considerations which has repeatedly been stressed in academic papers [11,50,58] and also holds for student theses. However, the observed power was higher than hypothetical power estimates for psychological studies [12].

## 4.2. Power and effect sizes

Notwithstanding the mentioned decent average power levels, the samples were not calibrated to detect expectable effect sizes. This results in the imperfect use of resources, either by running studies with samples too small to make precise effect estimate or by running studies with unnecessarily large samples. Of course, a large sample size generally is favourable to estimate effects of any size precisely, but in a situation where resources for human samples are limited (e.g. accessibility, remuneration), it is pivotal to calibrate the required sample size prior to data collection.

Additionally, *decent* power levels are not yet *perfect* and can be improved. There are two ways of doing this. First, students would simply have to increase their sample size. To keep the resource argument in mind, multiple students could be assigned one larger research topic, where data are collected in a team effort to achieve a larger sample, but each student has their own specific research question with independent hypotheses. Second, students could study research topics where larger effect sizes can be expected. This can be achieved by studying new facets of empirically established large effects and a stronger theoretical foundation of research questions. This could also reverse the decreasing trend of effect sizes over the years which was found in the exploratory analysis. To achieve this aim, it is pivotal for supervisors to make themselves familiar with power analysis for different research designs.

The exploratory analysis revealed that effect sizes decreased over time. This has also been documented for published papers—termed the decline effect—and has mostly been attributed to publication bias, where significant effects are more likely to be published first and subsequent work with moderate or no effect follow at a slower rate [84,85]. However, this explanation cannot be applied to master's theses where studies are mostly independent and publication bias can be ruled out. Motyl *et al.* [24] also find decreasing effects over the years and present further explanations that could potentially hold for the investigated theses. First, focal topics of interest may simply have changed to phenomena with smaller true effects over time. Second, in later years, samples might be more heterogeneous and/or data collection might be noisier which could increase measurement error (e.g. more online than laboratory studies).

## 4.3. Teaching

In terms of QRPs, we observed the reassuring result of no excessive signs of *p*-hacking or further data collection. While this is no proof for the complete absence of QPRs, it is suggestive of no systematic challenges and therefore a reassuring finding. Nevertheless, we feel confident that implementing key elements of open and transparent scientific practice in student supervision is important to foster critical reflection of the existing literature and rigorous research practices.

The observation that, in comparison with publishing researchers, students are less likely to engage in QRPs but are more likely to make mistakes calls for more thorough statistical and methodological training. One would probably expect higher variability in the reporting quality among students. After all, researchers are selected from the population of students who have an interest in going into academia. Another source of heterogeneity might be different motivations among students. On average, a student with a clear desire for a career outside of academia could potentially have less of a motivation to engage with all necessary details of an empirical project than a student who pursues a career within academia or has a strong personal interest in empiricism. Also note that we did not code many other important aspects of a scientific report (e.g. theoretical aspects, operationalizations or manipulations). During the coding process, it often seemed that the quality of such aspects leaves room for improvements. Some theses will therefore not reach a quality that would meet peer review requirements of journals, nor must that be expected from every single student. However, as instructors, it should be our aim to equip students with the required skills to conduct their own empirical projects and, maybe even more importantly, to understand and evaluate the quality and credibility of empirical work that is published. Additionally, if students are unable to meet certain standards due to limited resources, they should at least be aware of the imposed limitations and discuss these in their thesis.

Our call for teaching open and transparent research practices is also in line with recent demands by the student representatives of German psychology students. In a common position paper, they expressed the desire for open science as a topic in methods as well as basic and applied courses, preregistration of the final thesis and replication studies as thesis topic [86]. Accordingly, we encourage supervisors to implement critical reflections of research practices that cause increased false-positive rates and to foster open and reproducible research practices.

For instance, most study programmes should offer easy ways of implementing preregistration in empirical research seminars. Here, students usually first read the relevant literature, derive a research question and hypothesis and develop a research design. Students could either file a preregistration or write a short registered report at this stage. After data collection and analysis, the final report, study materials and data could be openly shared. This would also create a structure where students of future seminars could build on previous seminar papers, for instance, by directly replicating studies, advancing existing research designs or following meta-analytic research questions.

We are undoubtedly not the first to adopt or advocate teaching open and reproducible research practices. Teaching materials have been newly created or adapted by multiple teachers in recent years. Many such course syllabi have even been shared online [87]. We want to point out three particular projects that aim to facilitate open research practices of students through replications. First, there is the collaborative replication and education project (CREP; [88–90]). Here, undergraduate students conduct state-of-the-art replication studies of influential papers from the literature. Once a sufficient number of independent replications have been conducted, a multi-site meta-analysis is performed, and a replication report is submitted as a scientific paper. Second, it is worth mentioning the Hagen Cumulative Science Project [91], a large-scale replication project where students conduct replications as their bachelor's thesis. More than 80 replications have already been conducted. Third, a similar initiative exists at the University of Hong Kong where students conduct replications for their thesis [92]. In the first year, 45 replication projects were successfully conducted. One aim of the project is to ultimately publish the results in academic journals.

Such projects teach students how to conduct high-quality replications, follow clear research protocols, adhere to open research practices and, ideally, publish a scientific paper. Additionally, they foster advanced methodological and experimental skills and an awareness of the importance of cumulative science. Jekel *et al.* [91] argue that replication projects in teaching are not only beneficial for students, but also for teachers who can use their teaching obligations in a meaningful way by contributing to cumulative science. Finally, the scientific community at large benefits from a greater number of replication results. Such projects show that the benefit of conducting replications has been recognized

and put into action at multiple sites. All of these projects try to facilitate rigorously conducted, high-powered studies that add to the cumulative knowledge in our field, while at the same time teaching state-of-the-art research practices to students.

## 4.4. Conclusion

Clearly, our results cannot be generalized to other institutions, maybe not even to other supervisors within our own school. Nevertheless, we believe our results constitute an interesting perspective on a selected group's research practices and hope that the findings will stimulate further discussions about student supervision in the light of open and reproducible science. It remains to say that we encourage students to set high standards for their own research and strive to acquire the required knowledge and skills, and supervisors to be teachers and examples of rigorous and open research practices.

Data accessibility. Data and R code for this study are openly available and can be retrieved from https://osf.io/jwaqv/.

Authors' contributions. J.O. generated the idea for the study. J.O. and J.M. coded the data. J.O. analysed the data, J.M., M.V. and E.K. provided feedback on the analyses. J.O. wrote the first draft of the manuscript, all authors critically edited it. All authors approved the final submitted version of the manuscript.

Competing interests. The authors declare that there were no conflicts of interest with respect to the authorship or the publication of this article.

Funding. This research was supported by the Open Access Publishing Fund of the University of Vienna.

Acknowledgements. We thank Marcel van Assen, Michèle B. Nuijten and Hilde Augusteijn for providing helpful feedback on the data analyses and interpretation of results. Also, we thank Martina Brandtner for assisting in data coding.

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
