## [Reviewer comments · Royal Society Open Science]

Review History

RSOS-190738.R0 (Original submission)

Review form: Reviewer 1 (Anand Krishna)

Is the manuscript scientifically sound in its present form?

Yes

Are the interpretations and conclusions justified by the results?

No

Is the language acceptable?

Yes

Is it clear how to access all supporting data?

Yes

Do you have any ethical concerns with this paper?

No

Have you any concerns about statistical analyses in this paper?

No

Recommendation?

Major revision is needed (please make suggestions in comments)

Comments to the Author(s)

In this manuscript, the authors analyze a large set of student theses submitted to a specific supervisor, focusing on reporting of effect sizes and power analyses as well as on actual achieved power, reporting errors and p-curves.

I very much like the idea of using actual theses as a basis for analysis, as this is more objective than self-report measures and less susceptible to social desirability or self-serving biases. I also believe that this approach has good potential for providing a useful contribution to the literature. Unfortunately, I feel that the manuscript does not fulfil that potential in its current form and therefore recommend major revisions.

My major issue lies with the contribution to the literature this article wants to make. In its current form, the article presents mostly descriptive data from an admittedly nonrepresentative sample, then draws conclusions about what should be done in teaching methods. However, given the nonrepresentative sample, these conclusions and recommendations are unlikely to generalize. Furthermore, the recommendations the authors make are general enough that their link to this specific data is questionable. Under what plausible empirical circumstances would one NOT conclude that power analysis is useful to optimize sample sizes or that teaching open science is good? And if one could conclude this from any plausible state of data ("student theses were always well-powered" is not plausible), how does the data actually contribute to the discussion in the field? From this perspective, the article in its current form does not do enough to justify its publication. At the same time, the authors raise questions in their introduction that they do not address in their discussion, specifically about whether a lower rate of QRPs in student theses reflects the lack of a publish-or-perish culture for students.

Although the above may sound harsh, I believe this work has merit and could be revised to address these problems. Specifically, in my view, the authors need to systematically examine the differences between the environment in which the theses were written and the environment in which academic researchers work in greater detail (e.g. resources, degrees of freedom, motivational differences, choice of research topics...). The authors' detailed knowledge of the students' environment would be a unique benefit here. From such an examination, they could identify concrete factors that might contribute to differences in their measures for this sample compared to the broader academic field. This would allow them to fruitfully interpret their findings with regard to what these factors might influence - for example, why is the average power in this sample higher than in the rest of the field? - and what they apparently do not influence - for example, the average effect size in this sample is similar to other estimates in the field.

To extend the example, an interesting conclusion from this approach might be that students, though (trivially) less experienced in empirical research on average than professional researchers, do not produce weaker effects in their paradigms. If the average expected effect sizes of student theses in this sample also do not differ from those of professional researchers, this would imply that students are as good as professional researchers when it comes to producing effects in paradigms. This in turn would counter the argument advanced by some individuals that a

researcher's "skill" as an investigator is an important determinant of whether they can get an effect. This is just an example of the kind of conclusions that could be argued (though of course not conclusively demonstrated) based on the data.

If the authors can generate interesting conclusions from this data while remaining judicious in their interpretations and acknowledging the additional assumptions they make in doing so, I believe this paper is capable of making a useful contribution to the literature. However, my suggested method is just that - a suggestion. The authors may prefer other approaches, but I strongly recommend that they consider what the core contribution of their data actually is and make that clear, regardless of how they choose to do so.

I append several more specific points:

p. 5, last paragraph: "These reported QRP self-admission rates of students are lower than those of academic researchers (Fiedler & Schwarz, 2016; John et al., 2012)." - this is not quite correct. In fact, the student self-admission rates in Krishna & Peter are in many cases higher than those that Fiedler & Schwarz estimate for academic researchers (selectively reporting studies, excluding data after looking at results, rounding off p-values, failing to report all relevant conditions, falsifying data, falsely claiming that results are unaffected by demographics and stopping data collection after achieving desired result are all noticeably higher, claiming to have predicted an unexpected result is nominally higher). Only optional stopping and failing to report all relevant DVs are less common in students based on these numbers. The comparison to John et al. supports what the authors are saying here, but that is an inappropriate comparison, as it is between a lifetime incidence and a one-shot prevalence.

p. 7, first paragraph: while I appreciate and concur with the argument that motivated reasoning may play a role in QRP use, results from Krishna & Peter indicate that motivation to engage in their thesis is negatively correlated with self-reported QRP use. This would seem to point in the direction that on average, students would be motivated to conduct good research rather than to produce effects. This would add directly to this argument. However, on a macro level, I don't quite see what the "motivated reasoning" argument actually adds to the paper. The authors conclude this section with the hope that the existence of QRPs in master's theses will show whether these are due to socialization or to "publish-or-perish" - how is this linked to motivated reasoning? This needs to be reexamined.

With regard to the conclusion of the effects of publish-or-perish - if I understand the authors correctly, their argument can be stated thus: IF student theses are unaffected by publish-or-perish culture, YET they show QRP prevalences similar to academic researchers, THEN QRPs are likely due to common factors between students and researchers (such as socialization). To the degree that students show LESS QRPs than researchers, this difference is due to differential factors (such as publish-or-perish).

This argument is valid, but the premise that allows its application to publish-or-perish vs. socialization is questionable. The correlation between supervisor attitudes and students' QRP use has already been demonstrated (Krishna & Peter), so even if supervisor attitudes are in fact affected by unique factors, these factors will still likely indirectly impact student QRP use. Only if one assumes that the correlation between supervisor attitudes and student QRP use reflects NO causality of supervisor attitudes does the authors' argument hold, which is quite a strong assumption on its face. Put simply: if publish-or-perish affects supervisors and supervisors affect student theses, then publish-or-perish also affects student theses, albeit indirectly. This means that it is difficult to interpret QRP use levels in students as being uniquely associated with socialization. The strong conclusion the authors suggest here is therefore unwarranted, although a large difference in prevalence might provide some support (because it is less likely that a large difference is completely mediated through an indirect effect of supervisor attitudes).

That being said, this perspective of common factors vs. professional researcher-specific factors is useful and well-conceived. It might benefit the paper to utilize this in a different way: the fact that all of the evaluated theses come from a single university with at least some shared supervision is an opportunity, as it allows a systematic conceptual analysis of what differences in procedural degrees of freedom exist between these students and academic researchers. These differences can point us in the direction of what kind of circumstances might lead to what kind of QRPs. Even if we were to assume that students' motivations when conducting their thesis work are completely determined by their supervisors (an exaggerated position), we could still draw conclusions about external factors that predict QRP use. If the theses systematically differ from academic work in such factors (such as access to statistical software, time and resource constraints etc.), the authors are in a unique position to identify this and thereby investigate the possible role of such factors in any systematic differences. This is already touched upon in section 1.2, but the differences to professional academic research could be expanded upon. This work cannot DEFINITELY identify the causes of any observed differences between student theses and academic work, but it can certainly provide us with lots of insight into what POSSIBILITIES exist.

p.9-10, RQ1: It remains a little unclear what conclusions would be drawn from a difference between the average effect sizes determined in the literature and the average effect size in student theses. What would the average effect size in a student thesis tell us? Or is this purely for descriptive and informative purposes, without any desire to draw inferences about where any such difference might come from? Similar questions also apply to RQ 2 (p. 10-11) and RQ 3 (11-12).

p.13-14, RQ4: Given the focus of the introduction on QRPs, a line could be inserted here making explicit the connection of p-hacking to QRPs. This would help to relate the operationalized RQs to the initial theoretical questions.

p.14, RQ5: The authors suggest that students might engage in optional stopping under certain circumstances. Is this a possibility for students at the University of Vienna in terms of logistics/resources (e.g. lab space, participant incentives) and time?

p.25, second paragraph: "termed decline effect" should be "termed the decline effect"

p.24, last paragraph: "no prove of" should be "no proof of"

Review form: Reviewer 2 (Alyssa Counsell)

Is the manuscript scientifically sound in its present form?

Yes

Are the interpretations and conclusions justified by the results?

Yes

Is the language acceptable?

Yes

Is it clear how to access all supporting data?

Yes

Do you have any ethical concerns with this paper?

No

Have you any concerns about statistical analyses in this paper?

No

Recommendation?

Accept with minor revision (please list in comments)

Comments to the Author(s)

Please see attached file (Appendix A).

Decision letter (RSOS-190738.R0)

18-Jun-2019

Dear Dr Olsen,

The editors assigned to your paper ("Questionable research practices and statistical reporting in 250 economic psychology master's theses: A meta-research investigation") have now received comments from reviewers. We would like you to revise your paper in accordance with the referee and Associate Editor suggestions which can be found below (not including confidential reports to the Editor). Please note this decision does not guarantee eventual acceptance.

Please submit a copy of your revised paper before 11-Jul-2019. Please note that the revision deadline will expire at 00.00am on this date. If we do not hear from you within this time then it will be assumed that the paper has been withdrawn. In exceptional circumstances, extensions may be possible if agreed with the Editorial Office in advance. We do not allow multiple rounds of revision so we urge you to make every effort to fully address all of the comments at this stage. If deemed necessary by the Editors, your manuscript will be sent back to one or more of the original reviewers for assessment. If the original reviewers are not available, we may invite new reviewers.

If your study uses humans or animals please include details of the ethical approval received, including the name of the committee that granted approval. For human studies please also detail

whether informed consent was obtained. For field studies on animals please include details of all permissions, licences and/or approvals granted to carry out the fieldwork.

- Data accessibility

If you wish to submit your supporting data or code to Dryad (<http://datadryad.org/>), or modify your current submission to dryad, please use the following link:
<http://datadryad.org/submit?journalID=RSOS&manu=RSOS-190738>

- Competing interests

- Authors' contributions

- Acknowledgements

- Funding statement

Kind regards,
Alice Power
Editorial Coordinator

on behalf of Dr Essi Viding (Subject Editor)
openscience@royalsociety.org

Associate Editor's comments (Dr Essi Viding):

This is a very interesting and important article and I share the reviewers' enthusiasm for seeing this paper published in RSOS. I have acted as an Associate Editor on this paper and would recommend that the authors' take on board the reviewer comments and revise accordingly. I would also suggest the following:

I would encourage the authors to change the title, in its current format it seems unduly negative and has somewhat of a 'clickbait' quality. There is some evidence for improvement on research practices in the authors' analyses (even if many areas are still wanting) and I would prefer not to offer the most pessimistic headline with the potential to polarise and sully the field. Psychology has been leading the way in improving research practices and it would be my preference to frame this paper in that light. The paper itself is commendably balanced in its discussion of the topic (rather more so than the title would lead to expect). I would also encourage the authors to further flesh out the importance of rigorous training, this is a very important point, underscored by the data presented here. The authors might also want to consider discussing the fact that it is possible that not all student projects/supervision are of a quality that would meet peer review (nor might it be realistic to expect so, despite our best efforts to want to provide high quality training) and this is something that would be helpful to discuss (i.e. despite the ideal scenario, is it realistic to expect all student projects to reach a high standard, given the variable quality of students and also supervisors, who may in many cases not be research active?).

My suggestions are in addition to the reviewer ones, which will need to be considered.

I sincerely hope that the authors will resubmit the paper and respond to the editor and reviewer comments.

Kind Regards,
Essi Viding

Comments to Author:

Reviewers' Comments to Author:
Reviewer: 1

Comments to the Author(s)

In this manuscript, the authors analyze a large set of student theses submitted to a specific supervisor, focusing on reporting of effect sizes and power analyses as well as on actual achieved power, reporting errors and p-curves.

I very much like the idea of using actual theses as a basis for analysis, as this is more objective than self-report measures and less susceptible to social desirability or self-serving biases. I also believe that this approach has good potential for providing a useful contribution to the literature. Unfortunately, I feel that the manuscript does not fulfil that potential in its current form and therefore recommend major revisions.

My major issue lies with the contribution to the literature this article wants to make. In its current form, the article presents mostly descriptive data from an admittedly nonrepresentative sample, then draws conclusions about what should be done in teaching methods. However, given the nonrepresentative sample, these conclusions and recommendations are unlikely to generalize. Furthermore, the recommendations the authors make are general enough that their link to this specific data is questionable. Under what plausible empirical circumstances would one NOT conclude that power analysis is useful to optimize sample sizes or that teaching open science is good? And if one could conclude this from any plausible state of data ("student theses were always well-powered" is not plausible), how does the data actually contribute to the discussion in the field? From this perspective, the article in its current form does not do enough to justify its publication. At the same time, the authors raise questions in their introduction that they do not address in their discussion, specifically about whether a lower rate of QRPs in student theses reflects the lack of a publish-or-perish culture for students.

Although the above may sound harsh, I believe this work has merit and could be revised to address these problems. Specifically, in my view, the authors need to systematically examine the differences between the environment in which the theses were written and the environment in which academic researchers work in greater detail (e.g. resources, degrees of freedom, motivational differences, choice of research topics...). The authors' detailed knowledge of the students' environment would be a unique benefit here. From such an examination, they could identify concrete factors that might contribute to differences in their measures for this sample compared to the broader academic field. This would allow them to fruitfully interpret their findings with regard to what these factors might influence - for example, why is the average power in this sample higher than in the rest of the field? - and what they apparently do not influence - for example, the average effect size in this sample is similar to other estimates in the field.

To extend the example, an interesting conclusion from this approach might be that students, though (trivially) less experienced in empirical research on average than professional researchers, do not produce weaker effects in their paradigms. If the average expected effect sizes of student theses in this sample also do not differ from those of professional researchers, this would imply that students are as good as professional researchers when it comes to producing effects in paradigms. This in turn would counter the argument advanced by some individuals that a researcher's "skill" as an investigator is an important determinant of whether they can get an effect. This is just an example of the kind of conclusions that could be argued (though of course not conclusively demonstrated) based on the data.

If the authors can generate interesting conclusions from this data while remaining judicious in their interpretations and acknowledging the additional assumptions they make in doing so, I believe this paper is capable of making a useful contribution to the literature. However, my suggested method is just that - a suggestion. The authors may prefer other approaches, but I strongly recommend that they consider what the core contribution of their data actually is and make that clear, regardless of how they choose to do so.

I append several more specific points:

p. 5, last paragraph: "These reported QRP self-admission rates of students are lower than those of academic researchers (Fiedler & Schwarz, 2016; John et al., 2012)." - this is not quite correct. In fact, the student self-admission rates in Krishna & Peter are in many cases higher than those that Fiedler & Schwarz estimate for academic researchers (selectively reporting studies, excluding data after looking at results, rounding off p-values, failing to report all relevant conditions, falsifying data, falsely claiming that results are unaffected by demographics and stopping data

collection after achieving desired result are all noticeably higher, claiming to have predicted an unexpected result is nominally higher). Only optional stopping and failing to report all relevant DVs are less common in students based on these numbers. The comparison to John et al. supports what the authors are saying here, but that is an inappropriate comparison, as it is between a lifetime incidence and a one-shot prevalence.

p. 7, first paragraph: while I appreciate and concur with the argument that motivated reasoning may play a role in QRP use, results from Krishna & Peter indicate that motivation to engage in their thesis is negatively correlated with self-reported QRP use. This would seem to point in the direction that on average, students would be motivated to conduct good research rather than to produce effects. This would add directly to this argument. However, on a macro level, I don't quite see what the "motivated reasoning" argument actually adds to the paper. The authors conclude this section with the hope that the existence of QRPs in master's theses will show whether these are due to socialization or to "publish-or-perish" - how is this linked to motivated reasoning? This needs to be reexamined.

With regard to the conclusion of the effects of publish-or-perish - if I understand the authors correctly, their argument can be stated thus: IF student theses are unaffected by publish-or-perish culture, YET they show QRP prevalences similar to academic researchers, THEN QRPs are likely due to common factors between students and researchers (such as socialization). To the degree that students show LESS QRPs than researchers, this difference is due to differential factors (such as publish-or-perish).

This argument is valid, but the premise that allows its application to publish-or-perish vs. socialization is questionable. The correlation between supervisor attitudes and students' QRP use has already been demonstrated (Krishna & Peter), so even if supervisor attitudes are in fact affected by unique factors, these factors will still likely indirectly impact student QRP use. Only if one assumes that the correlation between supervisor attitudes and student QRP use reflects NO causality of supervisor attitudes does the authors' argument hold, which is quite a strong assumption on its face. Put simply: if publish-or-perish affects supervisors and supervisors affect student theses, then publish-or-perish also affects student theses, albeit indirectly. This means that it is difficult to interpret QRP use levels in students as being uniquely associated with socialization. The strong conclusion the authors suggest here is therefore unwarranted, although a large difference in prevalence might provide some support (because it is less likely that a large difference is completely mediated through an indirect effect of supervisor attitudes).

That being said, this perspective of common factors vs. professional researcher-specific factors is useful and well-conceived. It might benefit the paper to utilize this in a different way: the fact that all of the evaluated theses come from a single university with at least some shared supervision is an opportunity, as it allows a systematic conceptual analysis of what differences in procedural degrees of freedom exist between these students and academic researchers. These differences can point us in the direction of what kind of circumstances might lead to what kind of QRPs. Even if we were to assume that students' motivations when conducting their thesis work are completely determined by their supervisors (an exaggerated position), we could still draw conclusions about external factors that predict QRP use. If the theses systematically differ from academic work in such factors (such as access to statistical software, time and resource constraints etc.), the authors are in a unique position to identify this and thereby investigate the possible role of such factors in any systematic differences. This is already touched upon in section 1.2, but the differences to professional academic research could be expanded upon. This work cannot DEFINITELY identify the causes of any observed differences between student theses and academic work, but it can certainly provide us with lots of insight into what POSSIBILITIES exist.

p.9-10, RQ1: It remains a little unclear what conclusions would be drawn from a difference between the average effect sizes determined in the literature and the average effect size in student theses. What would the average effect size in a student thesis tell us? Or is this purely for descriptive and informative purposes, without any desire to draw inferences about where any

such difference might come from? Similar questions also apply to RQ 2 (p. 10-11) and RQ 3 (11-12).

p.13-14, RQ4: Given the focus of the introduction on QRPs, a line could be inserted here making explicit the connection of p-hacking to QRPs. This would help to relate the operationalized RQs to the initial theoretical questions.

p.14, RQ5: The authors suggest that students might engage in optional stopping under certain circumstances. Is this a possibility for students at the University of Vienna in terms of logistics/resources (e.g. lab space, participant incentives) and time?

p.25, second paragraph: "termed decline effect" should be "termed the decline effect"

p.24, last paragraph: "no prove of" should be "no proof of"

Reviewer: 2

Comments to the Author(s)
Please see attached file.

Author's Response to Decision Letter for (RSOS-190738.R0)

See Appendix B.

RSOS-190738.R1 (Revision)

Review form: Reviewer 1 (Anand Krishna)

Is the manuscript scientifically sound in its present form?

Yes

Are the interpretations and conclusions justified by the results?

No

Is the language acceptable?

Yes

Do you have any ethical concerns with this paper?

No

Have you any concerns about statistical analyses in this paper?

No

Recommendation?

Accept with minor revision (please list in comments)

Comments to the Author(s)

First, I would like to express my thanks for the authors' hard work in addressing both my comments and those of the editor and other reviewer. The manuscript is much improved and has better focus and conclusions. I have only one substantive point and one small issue that I feel should be addressed before publication.

The substantive point first: In their response to my comments and in the paper, the authors argue that students' QRP self-admission rates are lower than those of researchers (cf. p.5 footnote, p. 25 "One major difference [compared to publishing researchers] is the absence of substantial QRPs among students."). I believe this is an inappropriate conclusion as it stands, or at least one that deserves more scrutiny. I have no issue with the inference of QRP prevalence from p-curves - the limitations of this approach are stated and acknowledged appropriately by the researchers. My issue instead arises from the standard for comparison.

I would argue that using direct self-admission rates from John et al. and Fiedler & Schwarz is an inappropriate standard of comparison for these theses. John et al. and Fiedler & Schwarz both based their self-admission rates on lifetime incidences (questions of the type "Have you ever in your career..."). Therefore, an individual researcher would only have to engage in a QRP once over their multiple research projects to answer yes. Using these estimates as a probability basis for judging an individual study is therefore problematic. Fiedler & Schwarz discuss this issue in some detail when they explain their prevalence measures.

If the thesis studies discussed in this paper are sampled from the same distribution as professional researchers, then the (lower) prevalence estimates provided by Fiedler & Schwarz are better predictors of whether a given study will contain a QRP (if one accepts F&S' rationale for their prevalence calculations). Alternatively, the Krishna & Peter self-admission numbers also refer to one individual study rather than a career lifetime's work, so they would provide a good standard for comparison for student theses. As both of these comparison standards generally show lower percentages than John et al.'s study, it would also be more conservative for your work to discuss your own low QRP estimates with reference to them rather than to John et al.

I recognize that my argumentation here is not completely conclusive, but I believe this point has sufficient merit to be at least acknowledged in your discussion (even if you conclude that the John et al. estimates are more appropriate comparisons). The most important impact this would have on your interpretation of the results is to somewhat further qualify the conclusion that students engage in fewer QRPs than researchers. Note that I do still agree with this conclusion per se, as your p-curve evidence is fairly strong, I simply believe that without a brief discussion of this issue, it may be read as stronger than perhaps strictly warranted.

The small issue I have is with your motivated reasoning argument. Thanks for your comments on this, I now understand where you're coming from. The revisions to the paper have helped, but a slight tweak to explain the link between "students have to work hard on their thesis" and "students are motivated to reach positive results" might make it just a bit clearer (i.e. help the reader understand what you mean with students' "perceived truth", p.6, around l.42).

If these points are briefly addressed, I would gladly endorse the manuscript for publication!

I would also like to apologize to the authors if they found my reviews too wordy - in purely written communication with no back and forth, I tend to more exhaustive writing to make sure my meaning gets across clearly.

Signed

Anand Krishna

Review form: Reviewer 2 (Alyssa Counsell)

Is the manuscript scientifically sound in its present form?

Yes

Are the interpretations and conclusions justified by the results?

Yes

Is the language acceptable?

Yes

Do you have any ethical concerns with this paper?

No

Have you any concerns about statistical analyses in this paper?

No

Recommendation?

Accept as is

Comments to the Author(s)

The authors have done a good job addressing all of the reviewer comments. I have no further suggestions except for one very minor one regarding the title change: drop the "a description of" at the beginning because it doesn't help or change the meaning without it.

Decision letter (RSOS-190738.R1)

22-Nov-2019

Dear Dr Olsen:

On behalf of the Editors, I am pleased to inform you that your Manuscript RSOS-190738.R1 entitled "A description of research practices and statistical reporting quality in 250 economic psychology master's theses: A meta-research investigation" has been accepted for publication in Royal Society Open Science subject to minor revision in accordance with the referee suggestions. Please find the referees' comments at the end of this email.

The reviewers and Subject Editor have recommended publication, but also suggest some minor revisions to your manuscript. Therefore, I invite you to respond to the comments and revise your manuscript.

- **Ethics statement**

- Data accessibility

<http://datadryad.org/submit?journalID=RSOS&manu=RSOS-190738.R1>

- Competing interests

- Authors' contributions

- Acknowledgements

- Funding statement

Because the schedule for publication is very tight, it is a condition of publication that you submit the revised version of your manuscript before 01-Dec-2019. Please note that the revision deadline will expire at 00.00am on this date. If you do not think you will be able to meet this date please let me know immediately.

Kind regards,
Lianne Parkhouse
Editorial Coordinator
Royal Society Open Science
openscience@royalsociety.org

on behalf of Dr Essi Viding (Associate Editor) and Essi Viding (Subject Editor)
openscience@royalsociety.org

Associate Editor Comments to Author (Dr Essi Viding):

As you can see, both reviewers are happy to accept your paper for publication. I echo the reviewers and want to thank the authors for being responsive to feedback, which we all feel has

improved the paper and will hopefully increase its impact. I would recommend you address the remaining point from Reviewer 2, which should be simple to do. Then please submit your final version for publication.

Many thanks for submitting your work to RSOS!

Essi Viding

Reviewer comments to Author:

Reviewer: 1

Comments to the Author(s)

First, I would like to express my thanks for the authors' hard work in addressing both my comments and those of the editor and other reviewer. The manuscript is much improved and has better focus and conclusions. I have only one substantive point and one small issue that I feel should be addressed before publication.

The substantive point first: In their response to my comments and in the paper, the authors argue that students' QRP self-admission rates are lower than those of researchers (cf. p.5 footnote, p. 25 "One major difference [compared to publishing researchers] is the absence of substantial QRPs among students."). I believe this is an inappropriate conclusion as it stands, or at least one that deserves more scrutiny. I have no issue with the inference of QRP prevalence from p-curves - the limitations of this approach are stated and acknowledged appropriately by the researchers. My issue instead arises from the standard for comparison.

I would argue that using direct self-admission rates from John et al. and Fiedler & Schwarz is an inappropriate standard of comparison for these theses. John et al. and Fiedler & Schwarz both based their self-admission rates on lifetime incidences (questions of the type "Have you ever in your career..."). Therefore, an individual researcher would only have to engage in a QRP once over their multiple research projects to answer yes. Using these estimates as a probability basis for judging an individual study is therefore problematic. Fiedler & Schwarz discuss this issue in some detail when they explain their prevalence measures.

If the thesis studies discussed in this paper are sampled from the same distribution as professional researchers, then the (lower) prevalence estimates provided by Fiedler & Schwarz are better predictors of whether a given study will contain a QRP (if one accepts F&S' rationale for their prevalence calculations). Alternatively, the Krishna & Peter self-admission numbers also refer to one individual study rather than a career lifetime's work, so they would provide a good standard for comparison for student theses. As both of these comparison standards generally show lower percentages than John et al.'s study, it would also be more conservative for your work to discuss your own low QRP estimates with reference to them rather than to John et al.

I recognize that my argumentation here is not completely conclusive, but I believe this point has sufficient merit to be at least acknowledged in your discussion (even if you conclude that the John et al. estimates are more appropriate comparisons). The most important impact this would have on your interpretation of the results is to somewhat further qualify the conclusion that students engage in fewer QRPs than researchers. Note that I do still agree with this conclusion per se, as your p-curve evidence is fairly strong, I simply believe that without a brief discussion of this issue, it may be read as stronger than perhaps strictly warranted.

The small issue I have is with your motivated reasoning argument. Thanks for your comments on

this, I now understand where you're coming from. The revisions to the paper have helped, but a slight tweak to explain the link between "students have to work hard on their thesis" and "students are motivated to reach positive results" might make it just a bit clearer (i.e. help the reader understand what you mean with students' "perceived truth", p.6, around l.42).

If these points are briefly addressed, I would gladly endorse the manuscript for publication!

I would also like to apologize to the authors if they found my reviews too wordy - in purely written communication with no back and forth, I tend to more exhaustive writing to make sure my meaning gets across clearly.

Signed
Anand Krishna

Reviewer: 2
Comments to the Author(s)

The authors have done a good job addressing all of the reviewer comments. I have no further suggestions except for one very minor one regarding the title change: drop the "a description of" at the beginning because it doesn't help or change the meaning without it.

Author's Response to Decision Letter for (RSOS-190738.R1)

See Appendix C.

Decision letter (RSOS-190738.R2)

28-Nov-2019

Dear Dr Olsen,

It is a pleasure to accept your manuscript entitled "Research practices and statistical reporting quality in 250 economic psychology master's theses: A meta-research investigation" in its current form for publication in Royal Society Open Science. The comments of the reviewer(s) who reviewed your manuscript are included at the foot of this letter.

You can expect to receive a proof of your article in the near future. Please contact the editorial office (openscience_proofs@royalsociety.org) and the production office (openscience@royalsociety.org) to let us know if you are likely to be away from e-mail contact -- if

you are going to be away, please nominate a co-author (if available) to manage the proofing process, and ensure they are copied into your email to the journal.

on behalf of Prof Essi Viding (Subject Editor)
openscience@royalsociety.org

Appendix A

The authors conducted a comprehensive coding of particular research behaviours and practices in psychology Masters theses from the University of Vienna. Overall, I believe that the authors' approach and methods were scientifically sound and answered some interesting research questions. The paper was also well written and used accessible language. Their research questions followed from a well-organized introduction and literature review, and the methods and analyses further followed from these sections and were sufficiently justified. All of the materials were available on OSF and the R code was documented in a way that allowed me to see how they tested their hypotheses.

See my comments below for suggested changes or areas that require further clarification, evidence, or justification. Please note page numbers refer to the page X of 39 at the top of the document (not the page number at the bottom).

- 1) Why the focus on standardized effect sizes? Research question 1 asks “Do students report effect sizes and what is the average size of the focal effects”. Page 9, line 26 or so states “First, they allow science communication in a standardized metric; second, they are directly comparable”. This statement seems to suggest that unstandardized effects like raw mean differences or regression coefficients are not considered “effect sizes”. I also disagree that standardized effects are “directly comparable,” although this is a contentious point in the literature. Standardization involves removing different scales by based on the sample standard deviation – and different samples will necessarily have different standard deviations. Challenges arise when using different scales/methods to compare the same construct but in many cases the unstandardized effects are more meaningful and interpretable to researchers. Wilkinson and the task force (1999) give the example of mean difference on cigarettes smoked – a 7 cigarette difference is much easier to understand than a Cohen's d of .3 or something. My point is that I don't think standardized effects are better or necessarily more useful/comparable than unstandardized effects, and I certainly think it would be a mistake to say that students do not report effect sizes if they do not report a standardized one.

If you only want to focus on standardized effects, you should revise your research question because it is currently misleading to ask “do students report effect sizes” when what you really mean is “do students report standardized effect sizes”. This conflation is unfortunately prevalent in psychology, and I think contributes to misunderstandings about what an effect size is. In your results, you comment on proportions of standardized effect sizes and mention cases where with means and SDs reported, a standardized effect could be calculated, but the language around “effect size” could be improved since unstandardized effect sizes are perfectly valid too. If you haven't seen it yet, I'd recommend this article:

Pek, J., & Flora, D. B. (2018). Reporting effect sizes in original psychological research: A discussion and tutorial. *Psychological Methods*, 23(2), 208-225.
<http://dx.doi.org/10.1037/met0000126>

- 2) Page 19: Post-hoc power: I really like your plots and your 2nd approach to discussing power in the theses, but I would recommend either removing the post-hoc power analyses

and focus on the power at different effect sizes, or add in an important discussion about post-hoc power and why this is a terrible approach. As described in Cumming's *The New Statistics* book, post-hoc power results are "illegitimate". It is better to examine the power/sample size results associated with the smallest effect size of interest or a particular level of an effect size. There is a lot of variability around the effect observed, and because power is the probability of rejecting the null if there is an effect in the population, this probability is meaningless when the null has already been rejected. Your second approach is perfect – so I would prefer to just have that information in the paper.

- 3) Given that the sample only included Master's theses from the University of Vienna, I think more information about the group of students, department, etc. is warranted. You make the argument that students are less likely to succumb to publication bias because they do not publish their theses, but I don't necessarily think this is the case at most institutions. Students going on to the PhD level with interests in academia know about the "publish or perish" mentality so they are not necessarily less susceptible to engaging in questionable research practices than other groups of academics. This is an important point that requires a bit more discussion than is currently in the paper. I'm also curious whether you think that there would be differences between these results if PhD dissertations had been analyzed instead.

Minor points:

Page 18 line 39: I would like to see more descriptive statistics for the sample size reported in the theses. You currently provide the median N. How much variability is there? SD, range, etc would be useful.

Page 22: Why did you use Spearman's rank order correlation on log transformed N? Please provide a justification for this. Would the results differ from a non-transformed N?

Page 23: You make the argument that p value inconsistencies. "were constant over time" but what you really mean is that they are not statistically related to time. Failing to reject the null hypothesis does not mean that the null hypothesis is necessarily true. To properly assess stability over time you would need to use an equivalence test. I think I would prefer to see graphs with year on the x axis and reporting practices on the y axis for a better understanding of variation over time (may be nonlinear for example).

On the whole, I liked this paper and believe that, with some minor revisions, it could be publishable in Royal Society Open Science.

Dr. Alyssa Counsell
Department of Psychology
Ryerson University
Toronto, ON, Canada

Appendix B

Associate Editor's comments (Dr Essi Viding):

This is a very interesting and important article and I share the reviewers' enthusiasm for seeing this paper published in RSOS. I have acted as an Associate Editor on this paper and would recommend that the authors' take on board the reviewer comments and revise accordingly. I would also suggest the following:

First of all, we would like to thank you for handling our manuscript, providing valuable feedback, and giving us the opportunity of improving the manuscript.

I would encourage the authors to change the title, in its current format it seems unduly negative and has somewhat of a 'clickbait' quality. There is some evidence for improvement on research practices in the authors' analyses (even if many areas are still wanting) and I would prefer not to offer the most pessimistic headline with the potential to polarise and sully the field. Psychology has been leading the way in improving research practices and it would be my preference to frame this paper in that light. The paper itself is commendably balanced in its discussion of the topic (rather more so than the title would lead to expect).

Thank you for this comment. We never had the intention to create a pessimistic view on the topic and appreciate this feedback. We changed the title to be more accurate. We hope this change also addresses some general points raised by Reviewer 1. It now reflects the descriptive nature of the study.

I would also encourage the authors to further flesh out the importance of rigorous training, this is a very important point, underscored by the data presented here. The authors might also want to consider discussing the fact that it is possible that not all student projects/supervision are of a quality that would meet peer review (nor might it be realistic to expect so, despite our best efforts to want to provide high quality training) and this is something that would be helpful to discuss (i.e. despite the ideal scenario, is it realistic to expect all student projects to reach a high standard, given the variable quality of students and also supervisors, who may in many cases not be research active?).

We extended our discussion section on the importance of rigorous training and added some reflections about the expected variability in theses' quality in the introduction as well as discussion. We also revised the introduction to express that only few students actually go into academia after graduation, which might be one major source of heterogeneity.

My suggestions are in addition to the reviewer ones, which will need to be considered. I sincerely hope that the authors will resubmit the paper and respond to the editor and reviewer comments.

Kind Regards, Essi Viding

Reviewer: 1

In this manuscript, the authors analyze a large set of student theses submitted to a specific supervisor, focusing on reporting of effect sizes and power analyses as well as on actual achieved power, reporting errors and p-curves.

I very much like the idea of using actual theses as a basis for analysis, as this is more objective than self-report measures and less susceptible to social desirability or self-serving biases. I also believe that this approach has good potential for providing a useful contribution to the literature. Unfortunately, I feel that the manuscript does not fulfil that potential in its current form and therefore recommend major revisions.

My major issue lies with the contribution to the literature this article wants to make. In its current form, the article presents mostly descriptive data from an admittedly nonrepresentative sample, then draws conclusions about what should be done in teaching methods. However, given the nonrepresentative sample, these conclusions and recommendations are unlikely to generalize. Furthermore, the recommendations the authors make are general enough that their link to this specific data is questionable. Under what plausible empirical circumstances would one NOT conclude that power analysis is useful to optimize sample sizes or that teaching open science is good? And if one could conclude this from any plausible state of data ("student theses were always well-powered" is not plausible), how does the data actually contribute to the discussion in the field? From this perspective, the article in its current form does not do enough to justify its publication. At the same time, the authors raise questions in their introduction that they do not address in their discussion, specifically about whether a lower rate of QRPs in student theses reflects the lack of a publish-or-perish culture for students.

We would like to thank the reviewer for their feedback. The example the reviewer provides is true. There is no empirical circumstance that would let us conclude that power analyses are not useful. However, keeping to this example, the data we inspect can tell us whether students make use of power analysis to a poor versus satisfactory degree. It is quite clear that this rate is currently poor. Therefore, we believe that we can draw some inferences from our descriptive analysis about current prevalence rates and derive recommendations from these observations.

Studying the prevalence of certain practices, that have been voiced as new standards of good scientific practice (i.e., power analysis, reporting standardized effect sizes, error control/evidential value, outcome independent reporting), among a selected group of students can, in our view, add to the discussion in our field. Clearly, the study would be even more informative if we had a representative sample. This is a central limitation of the study.

Regarding the question of whether QRPs are more likely to stem from psychologists' statistical training or rather from a "publish-or-perish" culture, we do not claim that we are able to fully answer this question with our study. We state that the data "might shed some light" on this question. At the lowest level of inference, the study adds to the debate where QRPs stem from, without giving a final answer to the question raised. To address the reviewer's concern, we rephrased this part to "provide a preliminary indication". We also explain the rationale in more detail. Also, in the title and introduction we state that the aim of the study is of descriptive nature in order to learn "to what extent students' actual behavior reflects sacrifices of scientific rigor to achieve significant results, considerations about statistical power, and correct statistical reporting, when writing their final thesis". We do see this as our main objective in this study.

Although the above may sound harsh, I believe this work has merit and could be revised to address these problems. Specifically, in my view, the authors need to systematically examine the differences between the environment in which the theses were written and the environment in which academic researchers work in greater detail (e.g. resources, degrees of freedom, motivational differences, choice of research topics...). The authors' detailed knowledge of the students' environment would be a unique benefit here. From such an examination, they could identify concrete factors that might contribute to differences in their measures for this sample compared to the broader academic field. This would allow them to fruitfully interpret their findings with regard to what these factors might influence - for example, why is the average power in this sample higher than in the rest of the field? - and what they apparently do not influence - for example, the average effect size in this sample is similar to other estimates in the field.

We thank the reviewer for this comment. We extended section 1.2 and now describe the key differences between students and researchers. One challenge in interpreting our results as suggested

by the reviewer is that there are multiple structural differences between students and researchers that are confounded. Ultimately, we do not really find any indication of severe QRPs among students. This means that we cannot isolate which of these structural differences drives the observed difference in QRP prevalence. The main structural differences between students and researchers are that students (1) usually do not pursue a publication of their findings, (2) have fewer financial means, (3) often face date dependent time pressure, and (4) have less scientific experience. We believe the most influential structural difference is the first point, which is why we highlighted this difference so much in the previous version of the manuscript. We now present all these differences in the introduction and they are also reflected in the revised discussion of the manuscript.

Regarding directly observed behaviors (achieved effect sizes, achieved power, number of inconsistencies, extent of reporting errors), the reviewer is absolutely right. The initial manuscript lacked a discussion of why we observe differences/similarities. We revised the discussion accordingly.

To extend the example, an interesting conclusion from this approach might be that students, though (trivially) less experienced in empirical research on average than professional researchers, do not produce weaker effects in their paradigms. If the average expected effect sizes of student theses in this sample also do not differ from those of professional researchers, this would imply that students are as good as professional researchers when it comes to producing effects in paradigms. This in turn would counter the argument advanced by some individuals that a researcher's "skill" as an investigator is an important determinant of whether they can get an effect. This is just an example of the kind of conclusions that could be argued (though of course not conclusively demonstrated) based on the data. If the authors can generate interesting conclusions from this data while remaining judicious in their interpretations and acknowledging the additional assumptions they make in doing so, I believe this paper is capable of making a useful contribution to the literature. However, my suggested method is just that - a suggestion. The authors may prefer other approaches, but I strongly recommend that they consider what the core contribution of their data actually is and make that clear, regardless of how they choose to do so.

We thank the reviewer for suggesting this specific point as an example. As stated in the previous comment, we revised the discussion and, for each observed point, now speculate where differences/similarities might stem from.

I append several more specific points:

p. 5, last paragraph: "These reported QRP self-admission rates of students are lower than those of academic researchers (Fiedler & Schwarz, 2016; John et al., 2012)." - this is not quite correct. In fact, the student self-admission rates in Krishna & Peter are in many cases higher than those that Fiedler & Schwarz estimate for academic researchers (selectively reporting studies, excluding data after looking at results, rounding off p- values, failing to report all relevant conditions, falsifying data, falsely claiming that results are unaffected by demographics and stopping data collection after achieving desired result are all noticeably higher, claiming to have predicted an unexpected result is nominally higher). Only optional stopping and failing to report all relevant DVs are less common in students based on these numbers. The comparison to John et al. supports what the authors are saying here, but that is an inappropriate comparison, as it is between a lifetime incidence and a one-shot prevalence.

This is an important point and we rechecked the comparison. When Krishna and Peter state that "comparing our results to Fiedler & Schwarz' estimated prevalence rates of individual QRPs, most QRPs have similar self-reported prevalences", they base this on the comparison of their assessed one time *self-admission* rates against Fiedler & Schwarz' *prevalence* estimates (product of *self-admission* rate and *in what percentage of their publications*). We believe this comparison is not justified. One would have to compare students' *self-admission* rates against Fiedler & Schwarz' *self-admission* rates. The same is true for the comparison between Krishna and Peter's and John et al.'s data. This is the case because Krishna and Peter only provide *self-admission* rates, and no further prevalence estimates.

We plotted the self-admission rates from the three publications side-by-side. Note that for Fiedler and Schwarz and John et al. these numbers are rough estimates that were extracted from the plots, so minimal deviations from the real raw data are possible.

As can be seen, the self-admission rates are clearly lowest among students, which is what we are referring to in the manuscript: “These reported QRP self-admission rates of students are lower than those of academic researchers (Fiedler & Schwarz, 2016; John et al., 2012)”. To avoid misinterpretation of this statement, we added a footnote explaining how we arrived at this statement.

p. 7, first paragraph: while I appreciate and concur with the argument that motivated reasoning may play a role in QRP use, results from Krishna & Peter indicate that motivation to engage in their thesis is negatively correlated with self-reported QRP use. This would seem to point in the direction that on average, students would be motivated to conduct good research rather than to produce effects. This would add directly to this argument. However, on a macro level, I don't quite see what the "motivated reasoning" argument actually adds to the paper. The authors conclude this section with the hope that the existence of QRPs in master's theses will show whether these are due to socialization or to "publish-or-perish" - how is this linked to motivated reasoning? This needs to be reexamined. needs to be reexamined.

The link between motivated reasoning and the socialization vs. “publish-or-perish” research question (now mitigated) is as follows. The source of QRPs among publishing psychologists is often explained with the narrative that in the presence of publication bias there can be an imbalance between the need to produce a publishable paper and conducting accurate research. However, this “publish-or-perish” narrative does not apply to students. So the question is, what are similarities between students and researchers? If we were to observe similar QRP rates, then one would be inclined to conclude that it is their belief about research practices (i.e., having to produce significant results). But for students this is not linked to publishing, so it would be reasonable to assume that it is linked to their own motivation (one potential common factor between students and researchers). We revised this section and hope it is easier to follow our rationale now.

Yes, the results by Krishna and Peter (2018) suggest that motivation to work on the thesis is negatively related to QRP use. However, in our view, the motivation to work on one's thesis is a different construct than motivated reasoning, which is often an unconscious process. It is not clear whether motivated reasoning should be more prevalent among students who state that they are more motivated to work on their thesis. It is also plausible that the belief in statistically significant results as being scientifically more relevant is just as prevalent among less motivated students.

With regard to the conclusion of the effects of publish-or-perish - if I understand the authors correctly, their argument can be stated thus: IF student theses are unaffected by publish-or-perish culture, YET they show QRP prevalences similar to academic researchers, THEN QRPs are likely due to common factors between students and researchers (such as socialization). To the degree that students show LESS QRPs than researchers, this difference is due to differential factors (such as publish-or-perish). This argument is valid, but the premise that allows its application to publish-or-perish vs. socialization is questionable. The correlation between supervisor attitudes and students' QRP use has already

been demonstrated (Krishna & Peter), so even if supervisor attitudes are in fact affected by unique factors, these factors will still likely indirectly impact student QRP use. Only if one assumes that the correlation between supervisor attitudes and student QRP use reflects NO causality of supervisor attitudes does the authors' argument hold, which is quite a strong assumption on its face. Put simply: if publish-or-perish affects supervisors and supervisors affect student theses, then publish-or-perish also affects student theses, albeit indirectly. This means that it is difficult to interpret QRP use levels in students as being uniquely associated with socialization. The strong conclusion the authors suggest here is therefore unwarranted, although a large difference in prevalence might provide some support (because it is less likely that a large difference is completely mediated through an indirect effect of supervisor attitudes).

We thank the reviewer for these suggestions. Interestingly, what we find is exactly what is stated in the last sentence of the reviewer's comment, namely that there is a large difference in prevalence of QRPs between publishing psychologists and students. What we always had in mind was to say that this *might* provide some support for the "publish-or-perish" explanation (see previous comment above). So our cautious interpretation is that this could suggest that the difference in QRPs is due to external factors that only apply to researchers (where the discussion around the "publish-or-perish" mentality in science is an immediate point that comes to mind). We revised this section to express our caution even more. We would like to thank the reviewer for making us reflect on this.

We generally agree that one would expect an indirect effect of supervisors' practices on students. However, we believe that such an effect must be smaller than if there was a direct effect within students' environment that leads to QRPs. As stated before, we further mitigated the part on this research question and do not regard it as a focal topic of the manuscript. We hope this is now reflected in our revised introduction and discussion.

That being said, this perspective of common factors vs. professional researcher-specific factors is useful and well-conceived. It might benefit the paper to utilize this in a different way: the fact that all of the evaluated theses come from a single university with at least some shared supervision is an opportunity, as it allows a systematic conceptual analysis of what differences in procedural degrees of freedom exist between these students and academic researchers. These differences can point us in the direction of what kind of circumstances might lead to what kind of QRPs. Even if we were to assume that students' motivations when conducting their thesis work are completely determined by their supervisors (an exaggerated position), we could still draw conclusions about external factors that predict QRP use. If the theses systematically differ from academic work in such factors (such as access to statistical software, time and resource constraints etc.), the authors are in a unique position to identify this and thereby investigate the possible role of such factors in any systematic differences. This is already touched upon in section 1.2, but the differences to professional academic research could be expanded upon. This work cannot DEFINITELY identify the causes of any observed differences between student theses and academic work, but it can certainly provide us with lots of insight into what POSSIBILITIES exist.

This point seems similar to the second major point raised above. Please see the response to the comment in the beginning.

p.9-10, RQ1: It remains a little unclear what conclusions would be drawn from a difference between the average effect sizes determined in the literature and the average effect size in student theses. What would the average effect size in a student thesis tell us? Or is this purely for descriptive and informative purposes, without any desire to draw inferences about where any such difference might come from? Similar questions also apply to RQ 2 (p. 10-11) and RQ 3 (11-12).

The main aim of the study is of descriptive nature. We revised the title to express this. However, as also suggested above by the reviewer, the particular result of similar effect sizes could be used as an argument that psychologists with less experience are also able to produce effect sizes of similar size, which is now mentioned in the discussion.

p.13-14, RQ4: Given the focus of the introduction on QRPs, a line could be inserted here making explicit the connection of p-hacking to QRPs. This would help to relate the operationalized RQs to the initial theoretical questions.

This is a good point. We added a sentence about the link of these concepts.

p.14, RQ5: The authors suggest that students might engage in optional stopping under certain circumstances. Is this a possibility for students at the University of Vienna in terms of logistics/resources (e.g. lab space, participant incentives) and time?

Yes, this is possible, especially for online studies that use convenience sampling. Here students could easily peek at the data and continue data collection if the result of their analysis is insignificant. The same applies to paper-pencil data collections. For lab experiments, it would in-fact be more difficult given the involved logistics (e.g., lab space, participant pool), but not strictly impossible.

p.25, second paragraph: "termed decline effect" should be "termed the decline effect" p.24, last paragraph: "no prove of" should be "no proof of"

We thank the reviewer for spotting these errors which were corrected.

We would like to thank the reviewer for their extensive review. We appreciate the critical and detailed suggestions.

Reviewer: 2

The authors conducted a comprehensive coding of particular research behaviours and practices in psychology Masters theses from the University of Vienna. Overall, I believe that the authors' approach and methods were scientifically sound and answered some interesting research questions. The paper was also well written and used accessible language. Their research questions followed from a well-organized introduction and literature review, and the methods and analyses further followed from these sections and were sufficiently justified. All of the materials were available on OSF and the R code was documented in a way that allowed me to see how they tested their hypotheses. See my comments below for suggested changes or areas that require further clarification, evidence, or justification. Please note page numbers refer to the page X of 39 at the top of the document (not the page number at the bottom).

1) Why the focus on standardized effect sizes? Research question 1 asks "Do students report effect sizes and what is the average size of the focal effects". Page 9, line 26 or so states "First, they allow science communication in a standardized metric; second, they are directly comparable". This statement seems to suggest that unstandardized effects like raw mean differences or regression coefficients are not considered "effect sizes". I also disagree that standardized effects are "directly comparable," although this is a contentious point in the literature. Standardization involves removing different scales by based on the sample standard deviation – and different samples will necessarily have different standard deviations. Challenges arise when using different scales/methods to compare the same construct but in many cases the unstandardized effects are more meaningful and interpretable to researchers. Wilkinson and the task force (1999) give the example of mean difference on cigarettes smoked – a 7 cigarette difference is much easier to understand than a Cohen's d of .3 or something. My point is that I don't think standardized effects are better or necessarily more useful/comparable than unstandardized effects, and I certainly think it would be a mistake to say that students do not report effect sizes if they do not report a standardized one.

If you only want to focus on standardized effects, you should revise your research question because it is currently misleading to ask "do students report effect sizes" when what you really mean is "do students report standardized effect sizes". This conflation is unfortunately prevalent in psychology, and I think contributes to misunderstandings about what an effect size is. In your results, you comment on proportions of standardized effect sizes and mention cases where with means and SDs reported, a standardized effect could be calculated, but the language around "effect size" could be improved since unstandardized effect sizes are perfectly valid too. If you haven't seen it yet, I'd recommend this article: <http://dx.doi.org/10.1037/met0000126>

This is a good point. We changed the wording of the research question to be more in line with what we actually did in our analysis, namely reporting *standardized* effect sizes. We fully agree that unstandardized effect sizes are important and can sometimes be easier to understand. The reason why we focused exclusively on standardized effect sizes was that our aim was to also aggregate the estimates (also Research Question 1), which is not directly possible with unstandardized effect sizes. We also improved the language around "effect sizes" in the entire manuscript and now more explicitly state that we are referring to standardized effect sizes only.

2) Page 19: Post-hoc power: I really like your plots and your 2nd approach to discussing power in the theses, but I would recommend either removing the post-hoc power and focus on the power at different effect sizes, or add in an important discussion about post-hoc power and why this is a terrible approach. As described in Cumming's *The New Statistics* book, post-hoc power results are "illegitimate". It is better to examine the power/sample size results associated with the smallest effect size of interest or a particular level of an effect size. There is a lot of variability around the effect observed, and because power is the probability of rejecting the null if there is an effect in the population, this probability is meaningless when the null has already been rejected. Your second approach is perfect – so I would prefer to just have that information in the paper.

We would like to thank the reviewer for pointing this out. One of the reasons why we included the 2nd approach is exactly this criticism of using observed power (along with the danger of underestimation of effect sizes in case of the absence of publication bias). However, this criticism is usually raised against calculating the post-hoc power of single studies where a single effect size estimate is taken for the true population effect size. For sets of studies, post-hoc power is still often used. The following blog post provides a detailed explanation why it is justified in meta-research:

<https://replicationindex.com/tag/observed-power/>. In conclusion, it is stated that post-hoc power can be useful and unbiased the analysis of several studies as the aggregation across studies reduces random

sampling error of effect sizes. We added a note to address the raised concerns and justify the used method.

3) Given that the sample only included Master's theses from the University of Vienna, I think more information about the group of students, department, etc. is warranted. You make the argument that students are less likely to succumb to publication bias because they do not publish their theses, but I don't necessarily think this is the case at most institutions. Students going on to the PhD level with interests in academia know about the "publish or perish" mentality so they are not necessarily less susceptible to engaging in questionable research practices than other groups of academics. This is an important point that requires a bit more discussion than is currently in the paper. I'm also curious whether you think that there would be differences between these results if PhD dissertations had been analyzed instead.

This is a very important point. We extended section 1.2 of the introduction about the environment students write their theses in at our school. As suggested by data from 2007, only 4% of students later find a research position after receiving their master's degree.

Dissertations are paper-based in psychology at the University of Vienna. This means working on a PhD means having to produce publishable manuscripts. There are studies that have investigated differences between dissertations that were initially not published and their resulting journal publication: O'Boyle, E. H., Banks, G. C., & Gonzalez-Mule, E. (2017). The Chrysalis effect: How ugly initial results metamorphosize into beautiful articles. *Journal of Management*, 43, 376–399. Here the authors find fundamental differences between the dissertation and final journal paper.

Minor points:

Page 18 line 39: I would like to see more descriptive statistics for the sample size reported in the theses. You currently provide the median N. How much variability is there? SD, range, etc would be useful.

We added information on *M*, *SD*, range, and *IQR*.

Page 22: Why did you use Spearman's rank order correlation on log transformed N? Please provide a justification for this. Would the results differ from a non-transformed N?

We calculated the correlation using the data before any transformations `cor.test(dat1_nona_r$SampleSize, dat1_nona_r$Pos_max_r, method = "spearman")`. We only transformed the data for better visualization as there were large outliers in the sample size distribution. Generally, the transformation of data does not influence the Spearman correlation coefficient as the relative ranks remain the same.

In section 2.3 we falsely stated "[for the correlation] we used the logarithmically transformed sample size". We deleted this sentence, as the transformation is only relevant for the plot and not the correlation.

Page 23: You make the argument that p value inconsistencies. "were constant over time" but what you really mean is that they are not statistically related to time. Failing to reject the null hypothesis does not mean that the null hypothesis is necessarily true. To properly assess stability over time you would need to use an equivalence test. I think I would prefer to see graphs with year on the x axis and reporting practices on the y axis for a better understanding of variation over time (may be nonlinear for example).

This is a good point. We changed the exploratory claim in this section and provide a supplementary plot (see Figure S2 in the OSF repository; <https://osf.io/jwaqv/>). The relationships do seem reasonably linear (slight U-shape tendency for reporting effect sizes).

On the whole, I liked this paper and believe that, with some minor revisions, it could be publishable in Royal Society Open Science.

Dr. Alyssa Counsell Department of Psychology Ryerson University Toronto, ON, Canada

We would like to thank Dr. Counsell for her extensive review. We appreciate the critical and detailed suggestions.

Appendix C

As you can see, both reviewers are happy to accept your paper for publication. I echo the reviewers and want to thank the authors for being responsive to feedback, which we all feel has improved the paper and will hopefully increase its impact. I would recommend you address the remaining point from Reviewer 2, which should be simple to do. Then please submit your final version for publication.

Thank you very much.

As suggested by Reviewer 2, we changed the title and omitted the „A description of“ part.